# Top-Down Polyelectrolytes for Membrane-Based Post-Combustion CO_2_ Capture

**DOI:** 10.3390/molecules25020323

**Published:** 2020-01-13

**Authors:** Daria Nikolaeva, Patricia Luis

**Affiliations:** UCLouvain—IMMC, Materials & Process Engineering, Place Sainte Barbe 2, 1348 Louvain-la-Neuve, Belgium; patricia.luis@uclouvain.be

**Keywords:** polyelectrolytes, polymerised ionic liquids, gas separation, CO_2_ capture, flue gas, quaternisation, self-assembly

## Abstract

Polymer-based CO_2_ selective membranes offer an energy efficient method to separate CO_2_ from flue gas. ‘Top-down’ polyelectrolytes represent a particularly interesting class of polymer materials based on their vast synthetic flexibility, tuneable interaction with gas molecules, ease of processability into thin films, and commercial availability of precursors. Recent developments in their synthesis and processing are reviewed herein. The four main groups of post-synthetically modified polyelectrolytes discern ionised neutral polymers, cation and anion functionalised polymers, and methacrylate-derived polyelectrolytes. These polyelectrolytes differentiate according to the origin and chemical structure of the precursor polymer. Polyelectrolytes are mostly processed into thin-film composite (TFC) membranes using physical and chemical layer deposition techniques such as solvent-casting, Langmuir-Blodgett, Layer-by-Layer, and chemical grafting. While solvent-casting allows manufacturing commercially competitive TFC membranes, other methods should still mature to become cost-efficient for large-scale application. Many post-synthetically modified polyelectrolytes exhibit outstanding selectivity for CO_2_ and some overcome the Robeson plot for CO_2_/N_2_ separation. However, their CO_2_ permeance remain low with only grafted and solvent-casted films being able to approach the industrially relevant performance parameters. The development of polyelectrolyte-based membranes for CO_2_ separation should direct further efforts at promoting the CO_2_ transport rates while maintaining high selectivities with additional emphasis on environmentally sourced precursor polymers.

## 1. Introduction

The steep increase in anthropogenic greenhouse gas (GHG) emissions over the last century, i.e., carbon dioxide (CO_2_), methane (CH_4_), nitrous oxides (NO_x_), water vapour (H_2_O), etc. led to irreversible environmental changes observed today [1]. These changes were spurred by the intensified fossil fuels consumption to sustain fast industrial and economic growth. High fuel consumption rate accelerated the release and accumulation of carbon dioxide (CO_2_) in the atmosphere disrupting the natural cycle of carbon.

Carbon capture, sequestration, and use (CCSU) strategies promote the responsible usage of fossil fuel through minimisation and treatment of exhaust GHG. CCSU directly implies the necessity to capture CO_2_ on-site at the production facilities, i.e., power plants, and to store it in suitable locations or to valorise it as a valuable source [2]. The CO_2_ release into atmosphere ought to be restricted through use of pre-combustion, oxy-fuel combustion and post-combustion capture. The post-combustion CO_2_ capture offers simpler control strategy for emissions containing low CO_2_ content or flue gas (<15 mol %), as it enables a direct integration of a purification step in the existing plant facilities (retrofitting) [3].

Post-combustion CO_2_ capture balances on a justifiable comparison between the most advanced CO_2_ separation technologies, including physical adsorption, chemical absorption, membrane separation and cryogenic distillation (Figure 1) [4]. The three main parameters to assess are CO_2_ content in the feed stream, energy required per kilogram of CO_2_ captured, and the purity of the CO_2_ reach product stream (permeate). Although the first three technologies offer robust solutions already at low CO_2_ concentrations in the feed (1–15%) and achieving the threshold of 80% CO_2_ purity, their energy demands vary drastically (Figure 1) [5,6,7,8]. Cryogenic distillation becomes financially attractive only if the feed already contains more than 90% CO_2_ [9]. This makes its direct application for one-stage post-combustion CO_2_ capture impossible and renders the membrane-based CO_2_ separation an aspiring technology for innovative development and research.

Membrane gas separation offers several advantages over conventional CO_2_ capture technologies (i.e., amine scrubbing, regenerative solvents and cryogenic distillation), such as moderate energy requirements, as well as operational and maintenance simplicity [10]. However, the implementation of membrane-based systems for CO_2_ recovery from post-combustion flue gases is hindered by low partial pressure of CO_2_ in the feed [11]. To compete with more mature technologies, membrane systems with advanced separation characteristics ought to be developed [7].

Membranes for flue gas CO_2_ capture processes should have suitable combination of CO_2_ permeability and carbon dioxide versus nitrogen (CO_2_/N_2_) selectivity, be thermally and chemically stable, and exhibit no plasticisation and/or physical ageing, while maintaining low cost and ease of module manufacturing [12]. Currently polyimide membranes occupy a major position on the post-combustion membrane-based separation market due to robustness [12,13]. Versatile chemical structures of these polymers provides increased CO_2_ solubility in the polymer matrix, and tailored chain packing density ensures faster diffusion rates through the selective layer [14,15]. Many alternative membrane concepts emerged based on polymers of intrinsic micro-porosity [16,17,18], thermally-rearranged polymers [19,20,21,22], organic/inorganic hybrid materials (mixed matrix membranes) [23,24,25,26], or facilitated transport materials [27,28,29,30]. Their advantages and disadvantages in membrane-based CO_2_ capture are summarised in Table 1. Facilitated transport materials are often associated with the quaternary ammonium groups also present in polyelectrolytes [31,32]. However, the scope of using polyelectrolytes in membrane-based CO_2_ separation from flue gas is much broader.

Polyelectrolytes have started gaining broad attention of the membrane community since the first publications emerged in the end of the last century. Later the unprecedented success of the ionic liquids (IL) have shifted the interest towards the monomer polymerisation approach with polymerisable ionic liquids (PILs) coming to the front line of the research on CO_2_ capture from flue gas (Figure 2a) [33,34,35,36,37]. This interest arose following the reports of high CO_2_ sorption capacities of IL and desire to improve long-term stability performance of IL membranes through combining the polymer chain flexibility and robustness with the physico-chemical potential of IL. Several detailed reviews were published in the last few years extensively covering all aspects of PIL synthesis and processing [38,39,40,41,42]. Thus, this review will mainly focus on ‘top-down’ functionalisation of existing and commercially available polymer precursors yielding advanced polyelectrolyte materials for CO_2_ separation (Figure 2b).

In this work, we summarise and critically review available polymer-derived polyelectrolytes (PEs) and their separation performance in processed form as membranes for the post-combustion CO_2_ capture. We attempt to differentiate between ‘bottom-up’ PEs synthesised from monomers and ‘top-down’ PEs prepared by functionalisation of existing natural and synthetic polymers. Furthermore, this review suggests the possible directions for ‘top-down’ PEs development in CO_2_ gas separation membranes.

**Table 1 molecules-25-00323-t001:** Comparison of various materials for membrane-based CO_2_ capture available.

Material	Advantages	Disadvantages	Commercial Product	Supplier	$\alpha_{\mathrm{CO}/N2}$ [-]	$\Pi_{\mathrm{CO}2}$ [GPU]	Ref.
Polymers	Endurance	Low permeance	Cellulose acetate (CA)	Sigma Aldrich	32	80	[43]
	Processing simplicity	Plasticszation	Matrimid^®^ 5218 (polyimide (PI))	Huntsman	33	0.14	[44]
	High selectivity		Torlon^®^ (polyamide-imide (PAI))	Solvay	34 ^a^–25 ^b^	0.47 ^a^–0.84 ^b^	[45]
			Pebax^®^ MH1657 (polyamide-polyether blocks)	Arkema	53	0.79	[46]
			Polyactive^™^ (polyethylene oxide terephthalate/polybutylene terephthalate)	PolyVation	53 ^a^–56 ^b^	1.81 ^a^–3.90 ^b^	[46,47,48]
Polymer of intrinsic	Rigid structure	Moderate selectivity	PIM 1	N/A	25	>12,000 ^c^	[49]
microporosity (PIMs)	Low polymer chain packing density	Production cost					
	High gas permeance	Physical ageing and plasticisation					
Thermally rearranged polymers (TRP)	Processability into hollow-fibres	mechanical strength after heat treatment	Poly(benzoxazoles)	N/A	⩽30	⩽1300 Barrer ^d^	[20,21,22]
Mixed-matrix membranes (MMM)	Excellent separation performance	Membrane processing	Various	N/A	>50	⩽2000 Barrer ^d^	[14,23,24,26,50,51,52,53]
Facilitated transport membranes	High selectivity	Poor chemical stability of the carriers	Polaris^™^ (based on Pebax^®^	MTR Inc.	50	1000–2200	[54,55]
	High permeability	evaporation and degradation					
		short lifetime					

^a^ Dense thick film membrane with average selective layer thickness $\bar{\delta }=$100μm. ^b^ Thin film composite membrane with $\bar{\delta }<$1μm. ^c^ CO_2_ permeance is estimated base on CO_2_ permeability value obtained from a membrane with $\bar{\delta }<$1μm. ^d^ Permeability value $P_{CO2}$ is reported in Barrer as no selective layer thickness was reported by authors.

## 2. Polyelectrolytes: Definition

Modern definition of polyelectrolytes (PEs) according to various sources describes them as polymer chains with charged monomer units that can dissolve into a charged macroion and small counterions upon the PE dissolution in a polar solvent [56,57,58]. The PEs properties can be ascribed to three major categories: origin, matrix, and charge (Figure 3). The PEs origin associates with the source of the raw polymer precursor, where the molecules such as proteins, cellulose, and deoxyribonucleic acid (DNA) represent the natural PEs. Opposed, with the development of chemical synthesis and especially polymeric chemistry a large field of synthetic polyelectrolytes have emerged to accommodate the needs of petrochemical, pharmaceutical, water recovery, and other industries [59]. These synthetic routes may roughly be distinguished as ‘bottom-up’ by monomer polymerisation and ‘top-down’ by post-synthetic modification of neutral polymers.

The PEs’ matrix differentiates the chemical structure of polymer chains within the substance as linear, branched, and cross-linked. In addition, the polymer chains can comprise homo- and co-polymers, as well as polymer blends, and organic/inorganic composites. Of course the main characteristic of PEs remains the charge properties of the ionisable groups: ionic type, position, and strength. The ionic type determines the charge carried by the ionized polymer (positive, negative or both), and PEs are defined as cationic, anionic, and polyampholytes, respectively. The strength of the PE charge controls PEs behaviour in a polar solvent through electrostatic forces. These can be weak, making the PEs behaving like a normal neutral polymer, or strong, exhibiting electrostatic coupling, i.e., polystyrenesulphonate (PSS) and poly(diallyldimethylammonium) chloride (PDADMAC) [58].

## 3. Post-Synthetic Modification of Polyelectrolytes

The early development of polyelectrolytes relied on industrial producers who aim at customer satisfaction and commercial attractiveness of the product, which often limits the fundamental understanding of underlying chemical interactions and their influence on application performance [60]. Despite the long history of polyelectrolyte application, research interest strongly persists in polyelectrolyte synthesis focused on their properties transfer to functional polymers, including strong long-range interaction, ionic conductivity, and hydrophilicity [61]. Polymer precursors used for the post-synthetic modification of polyelectrolytes maybe grouped into three categories according to the charge presence and sign (Figure 4): uncharged polymers (a), anionic polyelectrolytes (b), and cationic polyelectrolyte (c). All of these groups include both natural and synthetic precursors.

The precursors chemical structure allows several pathways for its synthetic modification. Figure 5 presents the conceptual summary of the four most implemented methods: ionisation (a), quaternisation (b), sulphation (c), cross-linking (d).

### 3.1. Ionisation

Ionisation is achieved by simply dissolving the polymers with amine groups (i.e., chitosan, poly(etherimide), etc.) in water and adjusting the solution pH below 3.0 through acidification with hydrochloric acid (Figure 5a) [62,63]. Same procedure is used to prepare PE from polymer precursors containing carboxylic groups by basification with sodium hydroxide [64]. The PE ionisation is most successful at low concentration of precursor in aqueous solutions ranging between $10^{-2}$ and $2\times 10^{-4}$ molL^−1^ (moles of monomer unit) [62,65]. This technique offers a major advantage in the absence of any additional chemical transformations. The disadvantage may stem from the necessity to modify a non-charged polymer prior the ionisation to incorporate any additional functional groups. Thus, this approach prioritises the use of existing commercially available polymers and also providing a monetary incentive to limit the use of expensive and dangerous reagents.

### 3.2. Cation-Functionalised Polyelectrolytes

Prevailing cationic PEs incorporate quaternary ammonium groups that provide almost inexhaustible combinations of nitrogen substituents for tailor-made polymer synthesis [43,66,67,68,69,70]. In addition, quaternary ammonium-based PEs exhibit basic properties, are relatively stable and cost-efficient [61]. Other heteroions in the cationic moieties may also be used, i.e., phosphonium, sulphonium, boronium. However, their application in membrane-based CO_2_ capture remains limited [71].

The quaternisation reaction consists of two steps: imidisation and quaternisation (Figure 5b) often followed by an optional anion exchange. The flexibility of PEs chemical structure may strongly affect their CO_2_ separation performance in several ways: basicity, hydrophobicity, tacticity of the precursor polymer, formation of ionic clusters, and ionic cross-linking. The bearing pendant group allows the adjustment of polycation basicity based on their estimated pKa values [72]. Thus, the pendant basicity controls the kinetics of CO_2_ binding and release resulting in release inhibition if the basicity is too high and weak binding if the basicity is too low.

Polycations carrying aromatic groups show increased hydrophobic interactions and π-stacking that enable the formation of tighter PEs assemblies [64]. These interactions may further exhibit synergistic effect with electrostatic forces if polymer precursors have different tacticity [73]. Both parameters significantly affect the formation of PEs-based films and determine their CO_2_ separation performance.

The preference for quaternary ammonium-based PEs is often explained by their preferential affinity for CO_2_ over other gas molecules, as CO_2_ reversibly interacts with the ion pairs in the presence of water molecules [31,74]. The ionic nature of PEs enables hydration of ionic moieties in the resulting membranes. This property to bind water around the ionic group facilitates the reaction between H_2_O and permeating CO_2_ molecule leading to the formation of HCO_3_^−^, which has higher permeation rates [25]. The CO_2_ transport may be further facilitated by changing the type and quantity of ionic groups and by creation of dedicated ionic clusters [31].

The ability of quaternary ammonium groups to engage in electrostatic and hydrophobic interactions, as well as hydrogen-bonding expands their possibilities to form tighter PEs films [75]. This properties may be enhanced through ionic cross-linking, often referred to as ‘glueing together’ [76]. Ionic cross-linking promotes the stiffness of the PEs matrix through incorporation of several quaternary ammonium groups per PEs monomer and leads to increased CO_2_ selectivity [77].

### 3.3. Anion-Functionalised Polyelectrolytes

Post-synthetic functionalisation to attain tailor-made PEs with cationic pendants prevails in the field of polyelectrolyte development for membrane-based for CO_2_ capture. However, less prominent anion-functionalised PEs play a vital role in the membrane preparation and are generally derived by sulphonation [73,76]. This method requires initial incorporation of aminealcohol molecules in the polymer chain via covalent coupling. The hydroxyl of aminealcohol further participates in the sulphonation reaction with sulphur trioxide where pyrridine and dimethylformamide are the solvents (Figure 5c) [78,79].

Polyanions comprising sulphonate and sulphate anionic moieties (i.e., PSS, heparin, poly(vinyl sulphonate) salts) yield highly stable PE films through the formation of strong and irreversibly ionically cross-linked pairs [80]. While this makes them the go-to solution for PE membrane synthesis, expansion of polyanion pool enables broader choice for task specific PE implemented in CO_2_ capture [71]. For instance, phosphonium-based hydroxides and carbonate present a successful case of stable thin-films with reversible CO_2_ solubility. As such the provide a solid base for their application as CO_2_ selective TFC membranes [71]. Even less frequently used polyanions are ones incorporating phosphates and phosphonates which exhibit relatively low pKa values and are stable against hydrolysis of the anionic groups [61,71]. However, to the best of our knowledge, these polyanions have not found any application in membrane-based CO_2_ separation from flue gas yet.

### 3.4. Cross-Linked Methacrylate-Based Polyelectrolytes

Methacrylate-based polymers are advantageous materials for membrane-based CO_2_ capture as the amino groups present in the polymer chains might enhance the CO_2_ separation performance through week acid-base interactions with the permeating CO_2_ molecules. N,N-dimethylaminoethanol methacrylate (DMAEMA) monomers were used to prepare homo- and co-polymeric membranes via chemical grafting and co-polymerisation with acetonitrile, respectively [81,82]. Alternatively, the direct synthesis of DMAEMA homopolymer offers a more straightforward approach for further membrane manufacturing [83]. This approach also renders the polymer matrix more durable through a direct bulk cross-linking with p-Xylylene dichloride (Figure 5d) [83].

## 4. Membrane Preparation

The industrial application of membranes for fluid streams separation (liquid and gas) became possible with the invention of phase inversion process for the preparation of polymeric asymmetric membranes by Loeb and Sourirajan [84]. This process ensured the formation of a thin selective dense layer on the surface of the membrane supported by the bulk porous structure. The thinner selective layer provides the faster mass transfer of the desirable component across the layer which is described by solution-diffusion mechanism [85]. In this case, the flux across the membrane exhibits a trade-off behaviour with the selectivity, comprehensively represented by a Robeson plot for gas separation of CO_2_/N_2_ [86]. Vast efforts were directed at the development of alternative membranes to combat the trade-off behaviour yielding materials with alternative transport mechanisms: molecular sieving effect, facilitated transport, and various combinations of thereof [87]. PEs materials proved to be a suitable material to enhance facilitated transport of CO_2_ molecules from flue gas. The stable PE-based membranes for CO_2_ capture were prepared as thin dense films using four main methods: solvent-casting layer deposition, Langmuir–Blodgett (LB) process, layer-by-layer (LbL) assembly, and chemical grafting on/to the support.

### 4.1. Solvent-Casting

Although methods discussed earlier offer some clear advantages in PE thin-film composite membrane preparation, the solvent-casting technique remains most frequently used on industrial and innovative level. Solvent-casting includes physical layer deposition by casting knife, dip, spin, and spray coating methods using a PE solution (Figure 6). The general rule dictates that the solvent used for solution preparation should be chemically stable and have a high boiling point to ensure slow evaporation process. The resulting dense selective film will yield a homogeneous surface without defects. However, careful pre-selection of solvents should be made for, as reports suggest that N, N-dimethylformamide can engage in hydrogen-bonding with sulphonated groups present in PE chains [88,89].

Solvent casting was successfully implemented in preparation of thick dense films (> 20 micron) and thin-film composites with a selective film (< 20 micron) based on various ‘top-down’ PE and PE-based materials [43,68,69,70,71,83,90,91,92,93]. Still the mechanical properties of the PE chains control the mechanical stability of the resulting films [67]. The selective layer is often considerably thicker when compared to self-assembly and grafting methods for the PE layer deposition. However, the reported values are from the scientific literature and they usually maybe interpolated to thinner layers if industrial equipment is used for the solvent-casting method.

### 4.2. Langmuir-Blodgett Films

Langmuir and Blodgett developed the method of monomolecular layer transfer from the water surface to a solid transfer, where the amphiphilic molecules (e.g., polyelectrolytes) arrange themselves on the interface between the aqueous and air phase. This method comprises three stages. Firstly, the PE monolayer is formed on the surface of the aqueous phase and is compressed with a special barrier to attain an uninterrupted surface layer ‘Step 1’ in Figure 7a.

Secondly, the film is transferred to a solid support by immersing a substrate in the aqueous phase and then passing it through the PE monoloyer ‘Step2’ (Figure 7a). At this moment, PE molecules adhere to the surface of the substrate and create a closed PE film, where the strength of the adsorbed film depends on the surface pressure of LB film [94]. Lastly, the recurrent LB film deposition allows the formation of layered PE surface films with a defined architecture and thickness ‘Step 3’ (Figure 7a) [95] that can be evaluated by ellipsometry and atomic force microscopy techniques [96].

A compressed dry LB film layer forms a crystalline structure with the crystal configuration dependent on the chemical structure and composition of the PE [97]. The crystalline structure will affect the thickness of the layer as well which consequently may have an effect on the separation performance of the LB thin-film PE membranes. The large number of system parameters affecting the physico-chemical properties of the deposited PE film result in both advantages and disadvantages for their application in membrane-based CO_2_ capture [98]. Ultimately, the tailor-made PE synthesis combined with versatile application of LB films encouraged the preparation of multiple PE membranes for CO_2_ separation from flue gas streams [63,99,100,101,102].

Nevertheless, LB has not found broad application in membrane fabrication on industrial level due to the PE specificity and long process time. Moreover, the LB monolayer films exhibit compromised mechanical stability and robustness in spiked conditions [103].

### 4.3. Layer-by-Layer Assembly

Opposed to LB, another method to prepare thin-film dense membranes from PE represents layer-by-layer self-assembly (LbL) by adsorption. First reported by Kirkland and Iler in 1965, the LbL technique allows the deposition of multilayered films comprising alternating layers of positively and negatively charged PEs on a substrate surface (Figure 7 b) [104,105]. The LbL deposition process consists of three major steps with intermediate rinsing with de-ionised water. In ‘Step 1’ the substrate surface with ionic interacting sites (here anionic) is submerged in the PE solution containing oppositely charged species (here cationic PE molecules). Upon withdrawal from the solution, the PE chain ionically adhere to the charged substrate surface exposing their cationic groups on the film surface. In ‘Step 2’, the substrate with positively charged semi-layer is submerged in the PE solution containing oppositely charged species (here anionic). In ‘Step 3’, upon withdrawal from the solution of anionic PE, the PE chains ionically adhere to the positively charged sites on the semi-layer. This step completes the deposition of a single bi-layer, and the procedure may be repeated till the desired number of bi-layers or PE film thickness is achieved.

This method offers considerable processing advantages over LB films through simplified layer deposition procedure, as PE molecule monolayer management is not required [106]. In addition, it enables fast application of multiple layers in line if the industrial production is desired. However, the main LbL associated drawback draws on similar issue of LB. The LbL assembly success is strongly dependent on the conditions of the aqueous PE solutions (i.e., ionic strength, pH, concentration), substrate (i.e., surface charge, anchors number, surface roughness), and surrounding (i.e., temperature, humidity) [103,107]. Nevertheless, LbL PE film application finds broad research interest in the CO_2_ separation from flue gases as advanced separation is expected [99,100,101,108].

### 4.4. Chemical Grafting

Alternatively PE film may be attached through chemical grafting of PE chains on the surface of the substrate (Figure 8a). The surface attachment offers two ways of grafting based on the type of chemical transformation: grafting ‘to’ and grafting ‘from’ [109]. Grafting ‘to’ employs a covalent reaction between the surface groups and a pre-made PE chains [110]. Grafting ‘from’ builds the PE chains from surface by using initiators [111].

Grafting outplays other physical PE film deposition methods (e.g., LB, LbL) by combining superior mechanical and chemical stability with inexhaustible variety of synthetic functional groups that can be incorporated in the PE chains [109]. Grafting ‘to’ method offers particularly simple film formation procedure. However, some considerations should be taken to the required grafting densities as the steric hindrance or ‘crowding’ may render the substrate surface less reactive. Moreover, the PE layer film thickness is limited by the length of the single polymer chain and cannot compete with the self-assembly (LB and LbL techniques) which offer multiple layer deposition opportunities.

Opposed to grafting ‘to’, grafting or polymerisation ‘from’ offers more controlled surface functionalisation with PEs with molecular precision in density and thickness. Thus, the film thickness maybe increased by the chain length of PE being attached, which depends not only on polymerisation time but also on spatial arrangements of the functionalities attached [112]. While many reports focus on polymerisation from surfaces, atom-transfer radical deposition allows preserving functional groups during the PE deposition process. However, the control precision often negatively affects the polymerisation speed [99].

When compared to LB and LbL methods, only Bruening et al. reports on preparation of ‘graft from’ PE-based membranes for selective separation of CO_2_ from flue gas [99]. The thickness of a defect free poly(ethylene glycol dimethylacrylate) (PEGDMA) films deposited on PSS/PAH coated substrate was 50 nm. This is comparable to the self-assembly methods based on the physical surface phenomena discussed earlier. In addition, this method could be further expanded to accommodate some CO_2_ selective functionalities.

## 5. PE-Based Membranes Show High Selectivities for CO_2_ in Flue Gas Separation

The fast growing numbers of newly synthesised polymer materials prompted Robeson to propose an empirical boundary for their gas separation performance evaluation in terms of permeability and selectivity for the given gas [86,113]. The boundary for various gas pairs including CO_2_/N_2_ is based on theoretical limit associated with the kinetic diameter of gas molecules. While the Robeson plot provides a comprehensive representation of material properties in comparison to theoretically possible, a similar plot proposed by Merkel et al. compares the membrane performance to industrially relevant standard and best market case in terms of permeance and selectivity for the given gas [54].

This review attempts to show the prospective of PE application in membranes for CO_2_ separation from flue gas on industrial level. Thus, the practical solution would be to combine Robeson and Merkel plot in one. For this purpose, all the PE-based membranes separation parameters reported in the literature for this application and discussed in the previous sections were converted to CO_2_ permeance (GPU) and CO_2_/N_2_ selectivity (dimensionless). If a hypothetical layer of the PE material is assumed to have a thickness of 1 micron then Robeson and Merkel plots can be merged and the permeability value will match the permeance value. Further in the text it is referred to as a joint R-M plot.

Figure 9 positions all materials covered in this review of the R-M plot. The majority of the materials discussed are situated in the top half of the plot. The exceptions are the first attempts to produce PE based membranes for CO_2_ capture by LbL assembly that show no selectivity for CO_2_ [62,65]. This supposedly was due to minor defects present in the deposited PE films.

PE-based membranes prepared by LB and LbL methods shows outstanding selectivities for CO_2_ and are positioned in the top left plot quarter in the vicinity of the Robeson boundary (Figure 9 in green oval). Several exemplars even overcome the Robeson boundary suggesting further possibilities for the performance improvement. However, these membranes demonstrate low CO_2_ permeance that cannot compete with the industrial market cellulose acetate (CA)-based standard for natural gas processing. This presents a drawback for industrial application as the selective layer matrix is highly cross-linked ionically and restricts fast transport of CO_2_ molecules across the membrane.

The thin-film composite membranes and a few LB composite membranes appear in the middle area of the plot. They show moderate performance in terms of selectivities comparable or inferior to market standard. However, some of these membranes have considerably improved CO_2_ permeances, which brings them closed towards the desirable area on the R-M plot. Such performance characteristics together with the ease of solvent-casting production technology, broadly available on the membrane market, make they a suitable candidate for further industrial upscaling of PE-based membranes for CO_2_ separation from flue gas.

## 6. Commercialisation Potential and Feasibility Testing

### 6.1. Production Up-Scaling

Up-scaling into line production with automated operation will further depreciate the costs for PE-based TFC membranes. In Table 2, the estimated price of the in-lab synthesised TFC membrane is calculated based on the following four components used: a PP/PE non-woven support, PI porous layer, CA-based PE selective layer, and PDMS sealing layer [43,114].

The price per unit weight of the CA-PE is calculated for a two step reaction that was not specifically optimised for a large scale production. The CA-PE reagents cost should further drop once they are acquired from a wholesale supplier instead of from the small scale laboratory suppliers. Such lab-scale suppliers (e.g., Sigma Aldrich, TCI, Acros Organics) that usually trade reagents at five to ten times higher margins. When the selective layer thickness approaches the industrial average, one order of magnitude lower than the lab-scale TFC membrane, the final price per unit area for CA-PE TFC membrane should be approaching 5 euro m^−2^. Although this is more than 60 % higher than commercial CA acetate membranes traded on Alibaba.com, the PE-based membrane achieves a two-fold increase in flux while maintaining relatively constant selectivity in a wide range of CO_2_ partial pressures.

Reported membrane separation performance and rough price estimation for a membrane area unit allow the socio-economic analysis of the membrane-based CO_2_ capture employing post-synthetically modified PE TFC membrane.

### 6.2. Feasibility Testing in Industrially Relevant Conditions

The real separation performance of TFC membranes for CO_2_ capture is largely affected by the presence of trace compounds in the industrial feed streams, such as SO_2_, H_2_S, and N_2_O, etc. To the best of our knowledge, membrane separation performance in conditions mimicking the real flue gas purification with trace compounds has only been investigated briefly [116]. Such limited number of research attempts is often connected with high costs, time, and safety requirements. However, further investigations will enable environmentalists and, with time, the broader interested public to employ the PE-based membranes in industrial installations [114].

## 7. Conclusions and Perspectives

Advanced polymer materials for membrane-based CO_2_ separation can be of elemental value in overcoming societal environmental challenges posed by CO_2_ commissions. Chemical flexibility in the development of PE offers broad range of possibilities for tuning their performance through various interactions within the PE film and with permeating CO_2_ molecules. Four general methods (i.e., ionisation, quaternisation, sulphation, and cross-linking) and their variations provide multiple opportunities to process PE in the way suitable both for large scale production and niche application in CO_2_ separation. Performance of PE-based materials depends on the intramolecular and intermolecular interaction between PE chains and ions in particular. LB and LbL film deposition ensures exceptionally high CO_2_ selectivities and limited CO_2_ flux. Grafting and solvent-casting film deposition allow higher CO_2_ fluxes with reduction in CO_2_ selectivity. Importantly the PE-based membranes for CO_2_ separation from glue gas almost enter the area of industrially desirable performance parameters on R-M plot more advanced research in the fundamental understanding of their interaction kinetics with CO_2_ in the flue gas are required to design superior PE-based membranes.

The future directions for PE-based membranes for CO_2_ capture should address the architectural adjustments for LB and LbL deposition techniques to promote their industrialisation. This may be encouraged by the application of composites combining self-assembly layer deposition methods with alternative materials to improve the flux across the selective layer while maintaining high selectivities. Finally, the development of new polymer-derived PEs should focus on bio-based precursor materials for PEs, such as cellulose acetate, chitosan, lignosulphates, etc. to support the stride for more environmentally conscious manufacturing of membranes for CO_2_ capture.

## Figures and Tables

**Figure 1 molecules-25-00323-f001:**
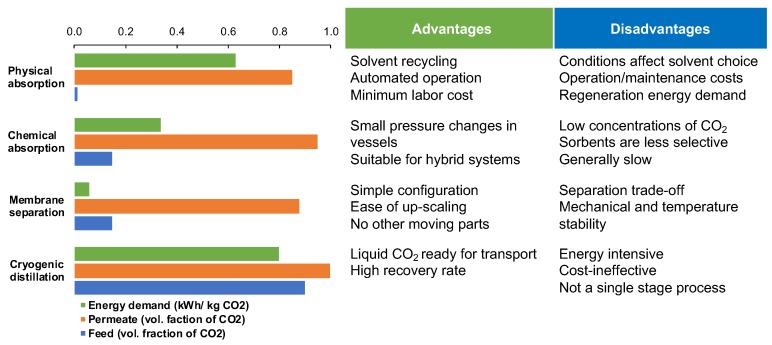
Comparison of existing CO_2_ capture technologies based on the energy demand for achieving certain volumetric ratio of CO_2_ in product stream and respective CO_2_ content in feed stream presented alongside the most common advantages and disadvantages of each process. The numerical data is collected across several publications and presented as reported [4,5,6,7,8,9].

**Figure 2 molecules-25-00323-f002:**
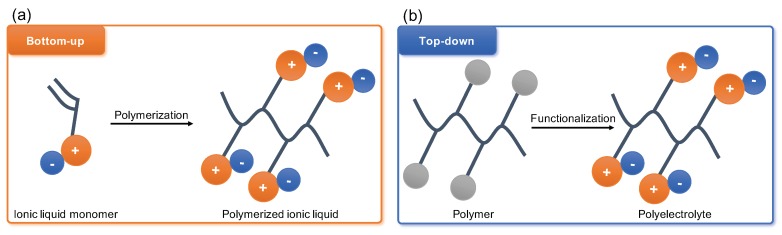
Crucial difference between polymerised ionic liquids (PILs) (**a**) and post-synthetic or ‘top-down’ modification of polyelectrolytes (**b**) stems from chemical origin of polymer chain bearing the ionic charges. In PILs it forms by polymerisation of ionic monomers, and in ‘top-down’ polyelectrolytes it is already present in parent material.

**Figure 3 molecules-25-00323-f003:**
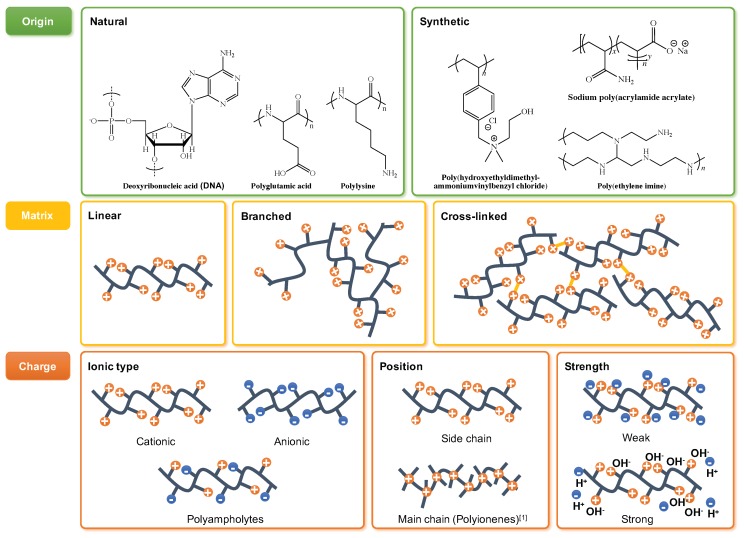
Origin, matrix, and charge in polyelectrolytes.

**Figure 4 molecules-25-00323-f004:**
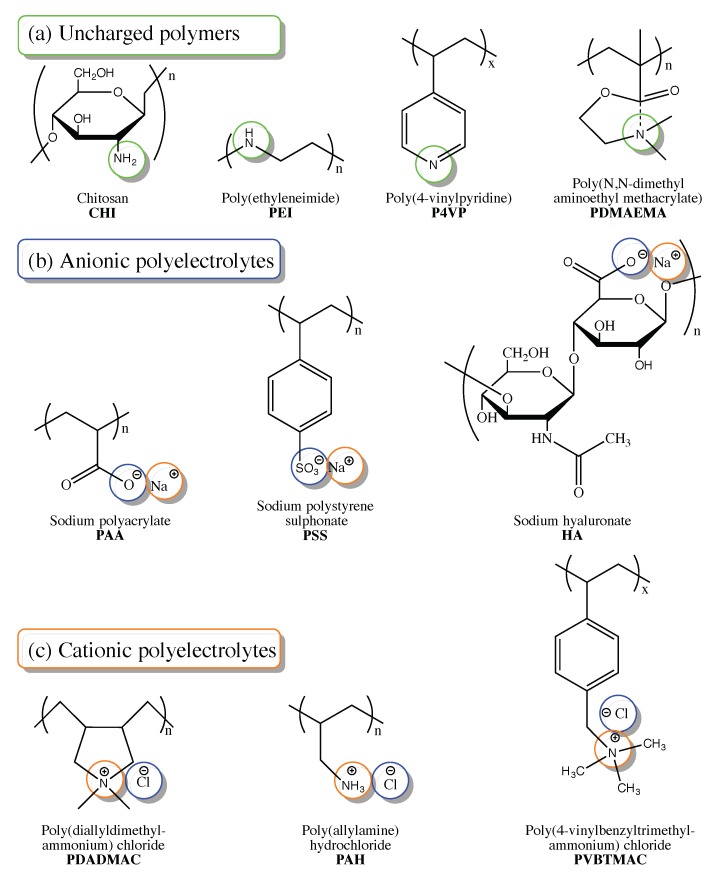
Chemical structures of polymers and polyelectrolytes used as precursors for post-synthetic material modification in membrane-based CO_2_ capture.

**Figure 5 molecules-25-00323-f005:**
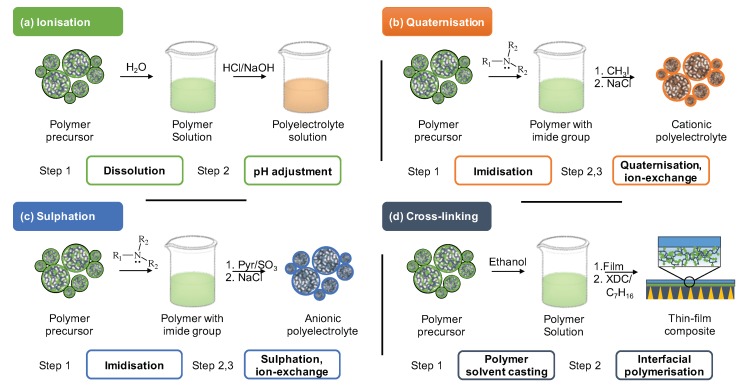
Major approaches for post-synthetic modification of polymer to produce ‘top-down’ polyelectrolytes include (**a**) ionisation, (**b**) quaternisation, (**c**) sulphation, (**d**) cross-linking. XDC abbreviates p-Xylylene dichloride.

**Figure 6 molecules-25-00323-f006:**
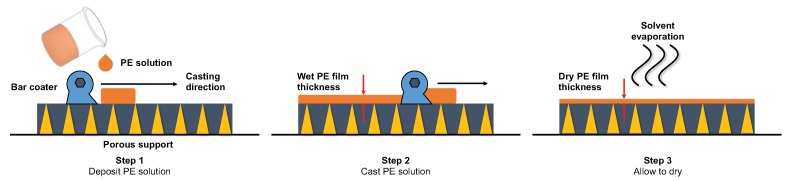
Physical layer deposition via solvent-casting method.

**Figure 7 molecules-25-00323-f007:**
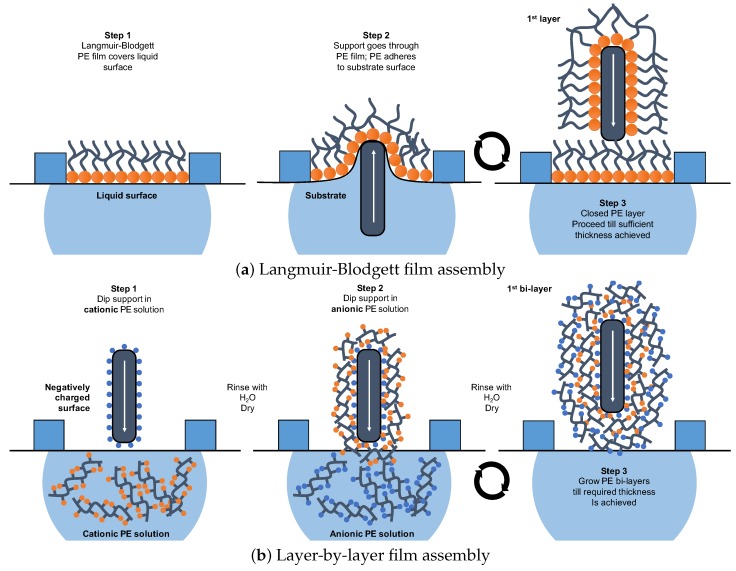
Physical layer deposition of polyelectrolyte films based on ionic interactions.

**Figure 8 molecules-25-00323-f008:**
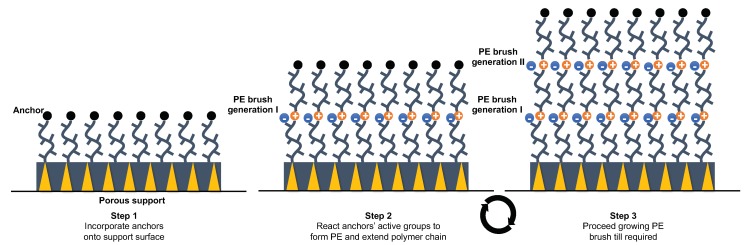
Film membranes Chemical layer deposition via grafting method

**Figure 9 molecules-25-00323-f009:**
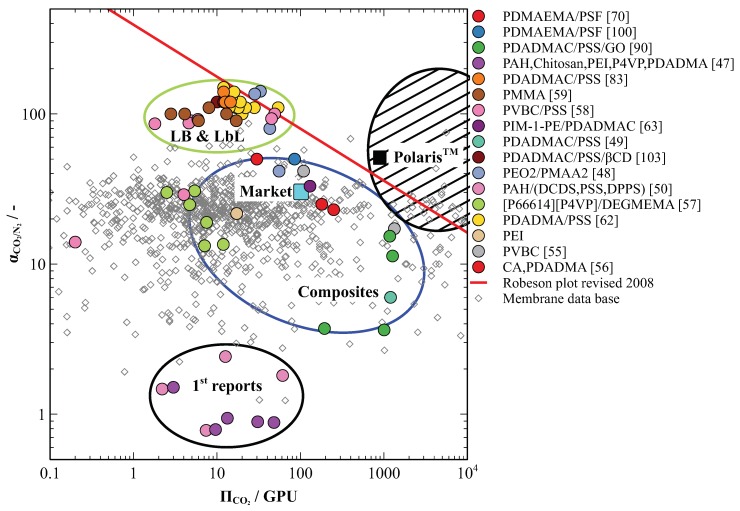
Comparison of PE-based membranes for CO_2_ separation from flue gas on the R-M plot where the value indicated with the cyan square represents the market standard for CO_2_ removal from natural gas and the value indicated with the black square represents the best performing membrane for CO_2_ capture available on the market. The striped area suggests the desirable separation performance of CO_2_ selective membranes for efficient capture from flue gas

**Table 2 molecules-25-00323-t002:** Cost estimation of TFC membrane with post-synthetically modified CA-derived PE selective layer based on the price of its comprising polymers. The table adapted from [114].

Component	ρ	δ	$\rho_{A}$ ^a^	Price per Unit Weight	Price per Unit Area ^b^
	(kg m ^−2^)	(μm)	(μm)	(euro kg^−1^)	(euro m^−2^)
PP/PE ^c^		100	0.07	3	0.21
PI ^c^	528 ^d^	50	0.0264	50	1.32
CA-PE ^e^	1480	5 (0.5) ^d^	0.0074	4745	35.113 (3.5) ^f^
PDMS ^c^	970	3 (0.3) ^d^	0.0029	5	0.0150 (0.0015) ^d^
**Total price per m^2^ of CA-PE TFC membrane**					37 (5)

^a^ Areal density of a two-dimensional object, such as sheet of paper. ^b^ Lab-made. ^c^ Prices from the wholesale supplier market Alibaba Group Holding Limited https://www.alibabagroup.com/en/global/home. ^d^ [115]. ^e^ CA-PE price per unit weight is estimated based on the synthesis procedure of P[CA][Tf_2_N] with reagents supplied by commercial laboratory suppliers, such as Sigma Aldrich, Acros, TCI, etc. [43]. ^f^ Idealised industrially produced membrane with thicknesses of CA-PE and PDMS layers ten times thinner than the lab-made TFC membrane.

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
