# Peer review of "Top-Down Polyelectrolytes for Membrane-Based Post-Combustion CO2 Capture"

_molecules, 2020, doi:10.3390/molecules25020323_

Round 1

Reviewer 1 Report

Manuscript No. 685374

The manuscript titled: Top-down polyelectrolytes for membrane-based post-combustion CO2 capture, looks interesting for the readers.  This paper is well written and structured which provides relevant insights. It deals with the review of the current developments works using polyelectrolyte-based membranes for CO2 separation. Moreover, the authors give an overview about the main membrane preparation procedures of such membranes. From my point overview, the article can be published in MOLECULES after some minor comments/suggestions.

Introduction: The authors should provide a table about the advantages and drawbacks of using this concept of membranes for the CO2 capture compared to other types of membranes (polymeric, MMMs, PIMs-based, thermally, facilitated-transport, amount others). The authors could provide small feedback about the techno-economic feasibility and potentiality of these membranes in CO2 capture towards real applications.

Author Response

First of all, we thank both reviewers for their critical analysis of our manuscript and for the useful suggestions for improvement. We have tried to implement all suggestions in the revised manuscript and, where available, we have illustrated our discussions with relevant additional references. All major changes are highlighted in the text (yellow) and a detailed response to all individual questions is also given below.

COMMENT 1

‘Introduction: The authors should provide a table about the advantages and drawbacks of using this concept of membranes for the CO2 capture compared to other types of membranes (polymeric, MMMs, PIMs-based, thermally, facilitated-transport, amount others).’

REPLY 1

Authors support the suggestion of the Reviewer # 1. The table # 1 comparing various types of membrane materials discussed in the manuscript was added to Introduction on page 3.

The following sentence was also added to announce the table in the text of the manuscript.

‘Advantages and disadvantages of aforementioned materials for membrane-based CO2 capture are summarised in Table 1 with relevant commercial products.’

Additional references were included to support the discussion of the existing literature in the Table 1.

Li, X.; Wang, M.; Wang, S.; Li, Y.; Jiang, Z.; Guo, R.; Wu, H.; Cao, X.; Yang, J.; Wang, B. Constructing CO2 transport passageways in Matrimid® membranes using nanohydrogels for efficient carbon capture. J. Memb. Sci. 2015, 474, 156–166. Kosuri, M.R.; Koros, W.J. Defect-free asymmetric hollow fiber membranes from Torlon®, a polyamide-imide polymer, for high-pressure CO2 separations. J. Memb. Sci. 2008, 320, 65–72. Lillepärg, J.; Georgopanos, P.; Shishatskiy, S. Stability of blended polymeric materials for CO2 separation. J. Memb. Sci. 2014, 467, 269–278. Maas, P.; Nauels, N.; Zhao, L.; Markewitz, P.; Scherer, V.; Modigell, M.; Stolten, D.; Hake, J.F. Energetic and economic evaluation of membrane-based carbon capture routes for power plant processes. Int. J. Greenh. Gas Control 2016, 44, 124–139. Sabetghadam, A.; Liu, X.; Gottmer, S.; Chu, L.; Gascon, J.; Kapteijn, F. Thin mixed matrix and dual layer membranes containing metal-organic framework nanosheets and PolyactiveTM for CO2 capture. J. Memb. Sci. 2019, 570–571, 226–235. Budd, P.M.; McKeown, N.B.; Ghanem, B.S.; Msayib, K.J.; Fritsch, D.; Starannikova, L.; Belov, N.; Sanfirova, O.; Yampolskii, Y.; Shantarovich, V. Gas permeation parameters and other physicochemical properties of a polymer of intrinsic microporosity: Polybenzodioxane PIM-1. J. Memb. Sci. 2008, 325, 851–860. Sen, D.; Kalipcilar, H.; Yilmaz, L. Development of zeolite filled polycarbonate mixed matrix gas separation membranes. Desalination 2006, 200, 222–224. Kentish, S.E.; Scholes, C.A.; Stevens, G.W. Carbon Dioxide Separation through Polymeric Membrane Systems for Flue Gas Applications. Recent Patents Chem. Eng. 2010, 1, 52–66. Bryan, N.; Lasseuguette, E.; Van Dalen, M.; Permogorov, N.; Amieiro, A.; Brandani, S.; Ferrari, M.C. Development of mixed matrix membranes containing zeolites for post-combustion carbon capture. Energy Procedia 2014, 63, 160–166. Li, S.; Jiang, X.; Sun, H.; He, S.; Zhang, L.; Shao, L. Mesoporous dendritic fibrous nanosilica (DFNS) stimulating mix matrix membranes towards superior CO2 capture. J. Memb. Sci. 2019, 586, 185–191. Jiang, X.; He, S.; Li, S.; Bai, Y.; Shao, L. Penetrating chains mimicking plant root branching to build mechanically robust, ultra-stable CO2-philic membranes for superior carbon capture. J. Mater. Chem. A 2019, 7, 16704–16711. Roussanaly, S.; Anantharaman, R.; Lindqvist, K.; Zhai, H.; Rubin, E. Membrane properties required for post-combustion CO2 capture at coal-fired power plants. J. Memb. Sci. 2016, 511, 250–264.

COMMENT 2

‘The authors could provide small feedback about the techno-economic feasibility and potentiality of these membranes in CO2 capture towards real applications.’ 

REPLY 2

Section 6 ‘Commercialisation potential and feasibility testing’ on pages 12 – 13 was added to provide an insight in techno-economic feasibility and future potential of PE-based TFC membranes.

Reviewer 2 Report

Polyelectrolytes have been widely studied in electrochemistry filed, while there are still less researches in gas separation. So It is meaningful to conduct such a review to conclude the polyelectrolyte membranes towards carbon capture in recent years. However, this manuscript still needs some revisions:

The authors should illustrate the benefit of polyelectrolyte membranes compared with other membrane materials in Introduction. Some traditional polymer materials can also realize great separation performance like PEO. It has been reported that the PEO-based membranes have realized about 2000-3000 Barrer in Journal of Membrane Science 586 (2019) 185 ; J. Mater. Chem. A 7 (2019) 16704. It’s confused that this manuscript didn’t give any living example in the argumentation. As far as I’m concerned, for a review, the readers might be more interested in some concrete instances rather than simple discussion, and the example can also make this review more convincing. The authors have missed one kind of polyelectrolyte membranes, which are made by simply adding salts into polymer matrix. Like Energy & Environmental Science 2014, 7, 1489; International Journal of Greenhouse Gas Control 2018, 78, 85. More recent references on this topic should be included and summerized.

Author Response

First of all, we thank both reviewers for their critical analysis of our manuscript and for the useful suggestions for improvement. We have tried to implement all suggestions in the revised manuscript and, where available, we have illustrated our discussions with relevant additional references. All major changes are highlighted in the text (yellow) and a detailed response to all individual questions is also given below.

Reviewer 2

Polyelectrolytes have been widely studied in electrochemistry filed, while there are still less researches in gas separation. So It is meaningful to conduct such a review to conclude the polyelectrolyte membranes towards carbon capture in recent years. However, this manuscript still needs some revisions:

COMMENT 1

‘The authors should illustrate the benefit of polyelectrolyte membranes compared with other membrane materials in Introduction. Some traditional polymer materials can also realize great separation performance like PEO. It has been reported that the PEO-based membranes have realized about 2000-3000 Barrer in Journal of Membrane Science 586 (2019) 185 ; J. Mater. Chem. A 7 (2019) 16704. It’s confused that this manuscript didn’t give any living example in the argumentation. As far as I’m concerned, for a review, the readers might be more interested in some concrete instances rather than simple discussion, and the example can also make this review more convincing.’

REPLY 1

This comment coincides with the comment of Reviewer # 1. The more detailed information should be consulted in the response to the comment # 1 of Reviewer # 1.

The table # 1 comparing various types of membrane materials discussed in the manuscript was added to Introduction on page 3.

The following sentence was also added to announce the table in the text of the manuscript.

‘Advantages and disadvantages of aforementioned materials for membrane-based CO2 capture are summarised in Table 1 with relevant commercial products.’

Additional references were included to support the discussion of the existing literature in the Table 1.

Li, X.; Wang, M.; Wang, S.; Li, Y.; Jiang, Z.; Guo, R.; Wu, H.; Cao, X.; Yang, J.; Wang, B. Constructing CO2 transport passageways in Matrimid® membranes using nanohydrogels for efficient carbon capture. J. Memb. Sci. 2015, 474, 156–166. Kosuri, M.R.; Koros, W.J. Defect-free asymmetric hollow fiber membranes from Torlon®, a polyamide-imide polymer, for high-pressure CO2 separations. J. Memb. Sci. 2008, 320, 65–72. Lillepärg, J.; Georgopanos, P.; Shishatskiy, S. Stability of blended polymeric materials for CO2 separation. J. Memb. Sci. 2014, 467, 269–278. Maas, P.; Nauels, N.; Zhao, L.; Markewitz, P.; Scherer, V.; Modigell, M.; Stolten, D.; Hake, J.F. Energetic and economic evaluation of membrane-based carbon capture routes for power plant processes. Int. J. Greenh. Gas Control 2016, 44, 124–139. Sabetghadam, A.; Liu, X.; Gottmer, S.; Chu, L.; Gascon, J.; Kapteijn, F. Thin mixed matrix and dual layer membranes containing metal-organic framework nanosheets and PolyactiveTM for CO2 capture. J. Memb. Sci. 2019, 570–571, 226–235. Budd, P.M.; McKeown, N.B.; Ghanem, B.S.; Msayib, K.J.; Fritsch, D.; Starannikova, L.; Belov, N.; Sanfirova, O.; Yampolskii, Y.; Shantarovich, V. Gas permeation parameters and other physicochemical properties of a polymer of intrinsic microporosity: Polybenzodioxane PIM-1. J. Memb. Sci. 2008, 325, 851–860. Sen, D.; Kalipcilar, H.; Yilmaz, L. Development of zeolite filled polycarbonate mixed matrix gas separation membranes. Desalination 2006, 200, 222–224. Kentish, S.E.; Scholes, C.A.; Stevens, G.W. Carbon Dioxide Separation through Polymeric Membrane Systems for Flue Gas Applications. Recent Patents Chem. Eng. 2010, 1, 52–66. Bryan, N.; Lasseuguette, E.; Van Dalen, M.; Permogorov, N.; Amieiro, A.; Brandani, S.; Ferrari, M.C. Development of mixed matrix membranes containing zeolites for post-combustion carbon capture. Energy Procedia 2014, 63, 160–166. Li, S.; Jiang, X.; Sun, H.; He, S.; Zhang, L.; Shao, L. Mesoporous dendritic fibrous nanosilica (DFNS) stimulating mix matrix membranes towards superior CO2 capture. J. Memb. Sci. 2019, 586, 185–191. Jiang, X.; He, S.; Li, S.; Bai, Y.; Shao, L. Penetrating chains mimicking plant root branching to build mechanically robust, ultra-stable CO2-philic membranes for superior carbon capture. J. Mater. Chem. A 2019, 7, 16704–16711. Roussanaly, S.; Anantharaman, R.; Lindqvist, K.; Zhai, H.; Rubin, E. Membrane properties required for post-combustion CO2 capture at coal-fired power plants. J. Memb. Sci. 2016, 511, 250–264.

The following recommended references on application of conventional polymers for mixed-matrix membranes were included in Introduction of page 3 as part of the comparative table # 1:

Li, S.; Jiang, X.; Sun, H.; He, S.; Zhang, L.; Shao, L. Mesoporous dendritic fibrous nanosilica (DFNS) stimulating mix matrix membranes towards superior CO2 capture. J. Memb. Sci. 2019, 586, 185 – 191. Jiang, X.; He, S.; Li, S.; Bai, Y.; Shao, L. Penetrating chains mimicking plant root branching to build mechanically robust, ultra-stable CO2-philic membranes for superior carbon capture. J. Mater. Chem. A 2019, 7, 16704 – 16711.

COMMENT 2

‘The authors have missed one kind of polyelectrolyte membranes, which are made by simply adding salts into polymer matrix. Like Energy & Environmental Science 2014, 7, 1489; International Journal of Greenhouse Gas Control 2018, 78, 85. More recent references on this topic should be included and summerized.’

REPLY 2

The authors agree with the reviewer that adding salts into polymer matrix may yield membranes with electrolyte properties. However, from the composition point of view these membranes would correspond to the definition of mixed matrix membranes where polymer would provide the matrix for charge carrying species, namely salts. According to the classical definition polyelectrolytes are polymers consisting of repeating monomer units bearing a charge. Thus, polymeric membranes with salts as mixed matrix membranes were outside the scope of this review, as it focused purely on post-synthetically modified polyelectrolytes.

After carefully reading the proposed references, the authors decided to include them as examples of alternative materials for CO2 capture in Introduction on page 2 under citation numbers 25 & 30.

Li, Y.; Xin, Q.; Wu, H.; Guo, R.; Tian, Z.; Liu, Y.; Wang, S.; He, G.; Pan, F.; Jiang, Z. Efficient CO2 capture by humidified polymer electrolyte membranes with tunable water state. Energy Environ. Sci. 2014, 7, 1489–1499. Zhang, H.; Tian, H.; Zhang, J.; Guo, R.; Li, X. Facilitated transport membranes with an amino acid salt for highly efficient CO2 separation. Int. J. Greenh. Gas Control 2018, 78, 85–93.
